**Data Availability Statement:** All data and coding files are available at https://github.com/UP-COP-SPSRA/ahsv_entry_assessment_zafcontrolledarea.

# An entry risk assessment of African horse sickness virus into the controlled area of South Africa through the legal movement of equids

John D. Grewar[1,2]*, Johann L. Kotze[1], Beverly J. Parker[2], Lesley S. van Helden[3], Camilla T. Weyer[2,4]

**1** Department of Production Animal Studies, University of Pretoria, Pretoria, Gauteng, South Africa, **2** South African Equine Health and Protocols NPC, Cape Town, Western Cape Province, South Africa, **3** Veterinary Services, Western Cape Department of Agriculture, Elsenburg, Western Cape Province, South Africa, **4** Department of Veterinary Tropical Diseases, University of Pretoria, Pretoria, Gauteng, South Africa

* jdgrewar@gmail.com

## Abstract

South Africa is endemic for African horse sickness (AHS), an important health and trade-sensitive disease of equids. The country is zoned with movement control measures facilitating an AHS-free controlled area in the south-west. Our objective was to quantitatively establish the risk of entry of AHS virus into the AHS controlled area through the legal movement of horses. Outcomes were subcategorised to evaluate movement pathway, temporal, and spatial differences in risk. A 'no-control' scenario allowed for evaluation of the impact of control measures. Using 2019 movement and AHS case data, and country-wide census data, a stochastic model was developed establishing local municipality level entry risk of AHSV at monthly intervals. These were aggregated to annual probability of entry. Sensitivity analysis evaluated model variables on their impact on the conditional means of the probability of entry. The median monthly probability of entry of AHSV into the controlled area of South Africa ranged from 0.75% (June) to 5.73% (February), with the annual median probability of entry estimated at 20.21% (95% CI: 15.89%-28.89%). The annual risk of AHSV entry compared well with the annual probability of introduction of AHS into the controlled area, which is ~10% based on the last 20 years of outbreak data. Direct non-quarantine movements made up most movements and accounted for most of the risk of entry. Spatial analysis showed that, even though reported case totals were zero throughout 2019 in the Western Cape, horses originating from this province still pose a risk that should not be ignored. Control measures decrease risk by a factor of 2.8 on an annual basis. Not only do the outcomes of this study inform domestic control, they can also be used for scientifically justified trade decision making, since in-country movement control forms a key component of export protocols.

**Funding:** This work is based on the research supported in part by the National Research Foundation of South Africa (Grant Number: 120319 – University of Pretoria Community of Practice in Sanitary and Phytosanitary Risk Assessment). The opinions, findings, conclusions and/or recommendations expressed in this publication are the authors alone. South African Equine Health and Protocols NPC (SAEHP) is a registered non-profit company in South Africa (registration number 2017/528099/08). It is privately funded, and the funders had no role in study design, data collection and analysis, decision to publish or preparation of the manuscript.

**Competing interests:** SAEHP functions through public private partnership agreement with the Western Cape Department of Agriculture: Veterinary Services. The ECOD case numbers used in this evaluation reflect the reporting system numbers and do not necessarily reflect the official totals as reported by South Africa's Veterinary Services. The authors have declared that no competing interests exist.

## Introduction

African horse sickness (AHS) is a disease of equines caused by an orbivirus, African horse sickness virus (AHSV). The disease occurs in *Equus caballus* (domestic horse), *Equus africanus asinus* (domestic donkey) and zebra (of which *Equus quagga* (common zebra) and *Equus zebra* (mountain zebra) are the most prevalent species in South Africa). Domestic horses are most susceptible to the disease and are the most likely of the hosts to show overt clinical signs [1, 2]. AHS is largely limited to the African continent where the disease is considered endemic in the sub-Saharan region [1]. A recent (2020) outbreak of AHS in Thailand, a country previously recognised as officially AHS free by the World Organisation for Animal Health (OIE), is an example of an outbreak outside Africa [3]. AHS is non-contagious and AHSV is vector borne, transmitted primarily by *Culicoides* midges. As such the mechanism of spread of the disease is intrinsically linked to the translocation of infected equids or the wind-dispersal of infected vectors [1]. The former mechanism of spread has been linked to outbreaks outside of Africa, with the Spanish outbreak in 1987 and the Thai outbreak of 2020 both linked to the movement of infected zebra [3, 4].

South Africa is considered endemic for AHS. However, the country is, to the best of our knowledge, the only country to date to maintain an explicit AHS controlled area that promotes trade. This zoning has been established based on the seasonal nature of the disease and host and environmental factors that create a risk differential between the AHS controlled area, in the south-western part of the country, and the larger endemic area in the rest of the country [5]. Movement control forms the primary basis for control of the potential entry of AHSV into the controlled area through the movement of equines [6, 7]. Base risk mitigation for the movement process includes veterinary health certification, vaccination against AHS and the issuance of permits from the veterinary authorities based on the status of AHSV circulation at origin. Depending on climatic conditions, that impacts both viral and vector replication and survival, the disease generally occurs between January (mid-summer) and May (late autumn) each year [8]. The outbreaks within the AHS controlled area, since 1999, have generally occurred slightly later (April–May) than the country-wide occurrence [9, 10]. South Africa has confirmed live-attenuated vaccine viral reassortment and/or reversion to virulence which has also resulted in outbreaks of disease [10].

In this study we estimate, using a stochastic model, the probability of entry [11] of AHSV into the AHS controlled area of South Africa through the legal movement of equids. Risk of further spread is not considered. The data set informing this analysis was obtained from movement and case data from the 2019 calendar year. The goal of the assessment was to identify, quantify and understand the temporal (monthly), spatial (local municipality) and movement type parameters associated with the risk of AHSV entry into the AHS controlled area of South Africa. A sensitivity analysis provides an evaluation of those parameters that result in entry risk uncertainty, while a 'what-if' scenario provides insight into the risk reduction of existing control measures when compared to an uncontrolled regulatory environment pertaining to movement. The benefits of quantifying entry risk lie not only domestically but can be used to justify movement control as one of the cornerstones that allow South Africa to trade horses internationally. Trade partners can make use of this information to evaluate AHS control measures and make informed decisions on trading with South Africa, particularly when such trade occurs from the AHS controlled area.

## Materials and methods

### Overview

A stochastic model was developed in R [12] with 10 000 iterations to establish the risk of entry of AHSV into the AHS controlled area, through the legal movement of equids, using 2019 case

and movement data. 95% credibility intervals (CI) reflected outcome uncertainty. The spatial unit of evaluation was at local municipality level which related to the level at which AHS case data was available. There are 234 local municipalities, which are public administrative regions, in South Africa. The temporal unit of measure was monthly. The probability of infection was based on reported AHS case data and was modulated by subclinical rates of disease along with the estimated population at risk within each area. Five distinct movement pathways were considered. For those movements through quarantine, we considered both the incubation period and viraemic periods of the virus within the host and considered the possibility that infected equids may have cleared their infection prior to final movement into the AHS controlled area. Also, for quarantine-based movements we considered two opportunities of infection where equids may be infected either prior to quarantine entry and/or during quarantine. Other than the increasing likelihood that equids may have cleared any pre-quarantine infection, we did not further consider quarantine risk mitigating benefits. Finally, equids moving from the same area through the same process were allocated the same parameter values for risk–i.e., they were treated as a cohort–for each iteration.

## Population data

A global distribution (GD) raster (0.0833 decimal degrees resolution) of population data was used to estimate spatial location of horses in South Africa [13]. South Africa's data in this raster was based on 2002 census figures provided by the then Directorate Veterinary Services. When aggregating these data to provincial level and comparing this to 2004 provincial census data (322 669 horses in South Africa—Department of Agriculture, Forestry and Fisheries (DAFF–unpublished data)) and to FAO country census totals for 2018 (325 805 horses in South Africa - http://www.fao.org/faostat/ accessed 1 Oct 2020), it was clear that the GD raster underestimated populations (248 437 horses in South Africa). For this analysis, however, the GD data provided the only dataset with the required resolution to estimate risk at a local municipality level. To account for this difference in population on aggregation, a factor differential was established at provincial level between the GD and the expected census based on 2004 data. This factor was then multiplied through each pixel in the GD raster for each province of South Africa using the raster package [14] in R, and then local municipality population estimates were obtained by aggregating pixel values for local municipality polygons. The final census dataset is available in the *data_census_rsa.csv* file in the dataset accompanying this manuscript.

## African horse sickness case data

AHS case data for 2019 was sourced from the existing Equine Cause of Disease (ECOD) surveillance program developed in collaboration between the South African Equine Veterinary Association (SAEVA) and the South African Equine Health and Protocols NPC (SAEHP). ECOD is an equine disease reporting platform for private equine veterinarians with a focus on AHS. Case data in ECOD is captured at local municipal area level. ECOD reporting is primarily based on laboratory testing reports from all laboratories testing for AHS in South Africa. Clinical cases epidemiologically linked to known cases can also be logged, but these are in the minority. Of all cases reported in 2019 (n = 613) 4.1% were considered clinical with no laboratory confirmation [15]. The case dataset is available in the *data_2019cases_ecod.csv* file in the dataset accompanying this manuscript.

## Equid movement data

Equid movement data were sourced from the permit system maintained by SAEHP on behalf of the Western Cape Department of Agriculture: Veterinary Services. Data were extracted for

the 2019 calendar year and included date of movement, number of equids associated with movement and point location of origin and destination for all permits issued where movement originated in the AHS endemic area of South Africa and the destination was in the AHS controlled area in either the AHS free, surveillance or protection zone. No donkeys were reported to have moved during 2019 into the AHS controlled area [16]. In the 2019 season one zebra moved into the AHS controlled area from the endemic area. The movement dataset is available in the *data_2019movements.csv* file in the dataset accompanying this manuscript.

**Movement patterns.** Permit based movements from the AHS endemic area of South Africa occur in five different pathways, a summary of which is shown in Table 1 with a spatial depiction of all movements shown in Fig 1. The first pathway, *Standard direct movements*, have horses originating in areas of the country classified as low AHS risk at the time of movement by the local government veterinarian. No quarantine at any point is required although compliant AHS vaccination certification is required and a pre-movement health check and certification by a private veterinarian must be performed. Quarantine associated movements are those classified as higher AHS risk at origin than is acceptable and direct movement is not permitted. Movements via quarantine are sub-classified by associated species, location, and level of vector-protection. The second pathway, *Standard stop-over quarantine (SOQ) movements*, are those associated with SOQ facilities that are in low-AHS risk locations in South Africa's AHS endemic area. Horses move from their high-risk areas into these facilities before moving into the AHS controlled area. While pre-quarantine AHS testing using a PCR test is not obligatory, this is generally undertaken. For the purposes of this analysis, and to remain as conservative as possible, this testing step is not assumed. A minimum of 14 days quarantine is undertaken. Vector mitigation measures include stabling of horses from two hours before sunset to two hours after sunrise and the application of insecticide and repellent twice daily. A negative AHSV PCR test is required prior to exit from quarantine. The third pathway, *Zebra quarantine (ZebraQ)*, is the movement process developed specifically for zebra moving into the AHS controlled area from the endemic area. These movements are only permitted between July and September (winter) to further mitigate AHS risk. Due to the subclinical nature of AHS in this species, a pre-movement health certification is not required; however, zebra require both a pre- and post-quarantine negative AHSV PCR test prior to movement. The pre-movement quarantine period for zebra is 21 days. The fourth pathway, *Vector-protected Quarantine at Origin (VPQO)*, constitutes movements from registered facilities that are based in high-AHS risk areas. These movements require the highest level of AHS risk mitigation and include both pre- and post-quarantine negative AHSV PCR testing. Horses are housed in

**Table 1. Movement pathways: Processes and associated risk classification.**

| Movement class | Origin area AHS risk classification | Pre-movement health certification | Quarantine based movement | Quarantine location AHS risk classification | Pre-quarantine PCR? | Post quarantine PCR? | Number (proportion) of equids moving in 2019 |
|---|---|---|---|---|---|---|---|
| 1. Standard direct | Low | Yes | No | N/A | N/A | N/A | 4099 (0.93) |
| 2. Standard stop-over quarantine (*SOQ*) | High[a] | Yes | Yes | Low | Not obligatory | Yes | 262(0.06) |
| 3. Zebra (*ZebraQ*) | Low[a] | No | Yes | Low | Yes | Yes | 1 (0) |
| 4. Vector protected quarantine (*VPQO*) | High[a] | Yes | Yes | High | Yes | Yes | 21 (0) |
| 5. Vector protected stop-over quarantine in AHS controlled area (*VPSOQ*) | High | Yes | Yes | Negligible (AHS controlled area) | Yes | Yes | 36 (0.01) |

[a]Exact area of origin not captured.

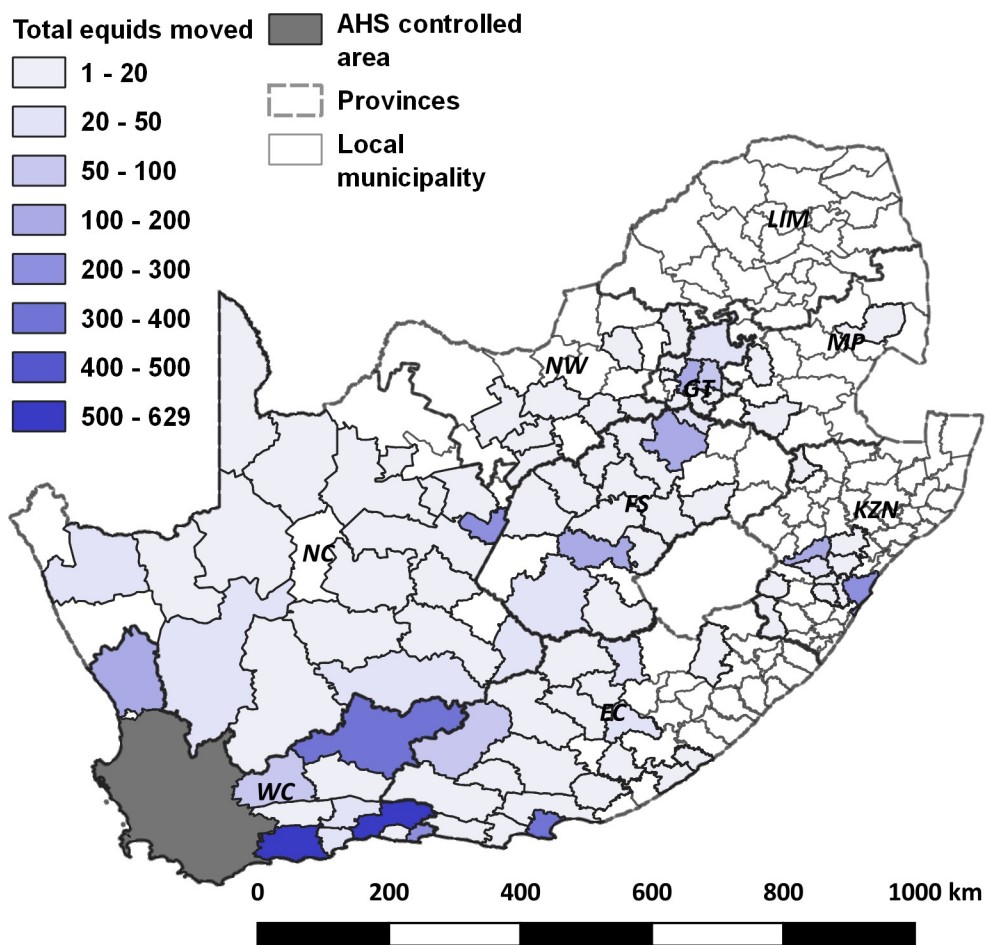

**Fig 1. Origin and total of all equid movements, by local municipality, into the African horse sickness controlled area (grey area in south-west part of the country) in 2019.** Provinces of the country are shown by dashed lines and are labelled. WC = Western Cape; EC = Eastern Cape; KZN = KwaZulu-Natal; MP = Mpumalanga; LIM = Limpopo; NW = North West; GT = Gauteng; FS = Free State; NC = Northern Cape.

vector-proof double door entry/exit barns except for a limited period between 10:00–15:00 daily when they are allowed access to non-vector protected camps for exercise and welfare reasons. Insecticides and insect repellents are applied to all horses prior to exit and re-entry into facilities. Finally, the fifth pathway, *Vector-protected SOQ (VPSOQ),* is a combination of standard SOQ and VPQO where the stop-over facility is vector-protected and is situated in the AHS protection zone of the AHS controlled area. The same risk mitigation measures are in place as for VPQO movements except that the facility is in a negligible risk location for potential infection of quarantined horses.

## Model parameters and processes

**Subclinical rate.** The subclinical rate of infection ($P_{subclin}$) for AHS cases influences both the probability of infection (since reported cases in the endemic area are generally through passive surveillance) and the probability that infected horses will be detected by private veterinarians during pre-movement health certification ($P_{detect}$). $P_{subclin}$ was based on the clinical case proportions observed in randomly selected outbreaks in the AHS controlled area where sub-clinical cases of the disease have been previously described (2011, 2014 and 2016 [17]). In

short, a *Beta*(*subclin_i*+1, *n_i*−*subclin_i*+1) distribution was used using an uninformed prior Bayesian estimate, where *subclin* subclinical cases were observed in outbreak *i* of a total of *n* outbreak cases.

**Probability of infection.** Probability of infection (*Pinf*) at origin was based on reported cases against an underlying population at risk for each local municipality for each month of the year. Case totals were amplified by a randomly selected subclinical rate per iteration. Where zero cases were reported for a month in a local municipality; and where subsequent movements of equids occurred, a more general estimate of *Pinf* was made, similarly to the methodology of de Vos et al. [18]. In short, a maximum cumulative incidence for that month for the AHS endemic area (population at risk of 311 433 horses) was established (since we know that cases did not occur within the AHS controlled area during 2019 based on OIE reporting (www.oie.int/wahis_2/public/wahid.php/Wahidhome/Home - accessed October 2020). For both local municipality-specific and general *Pinf* an uninformed prior Bayesian estimate was used. There were cases where, due to low numbers of horses in certain areas, the case total modulated by subclinical rate resulted in a total which exceeded the underlying population at risk. In these cases, *Pinf* was arbitrarily set to 0.5.

**Probability of detection.** All movement pathways, except for zebra movements, require health certification by a private veterinarian prior to movement. This health certification is specifically focussed on AHS detection and is certified by the veterinarian in each horse's passport. This certification is also dependent on the veterinarian's knowledge of the risk of AHS in the immediate area surrounding the movement origin. Clinical signs of AHS are not pathognomonic and, as this may result in some bias towards certifying AHS freedom clinically, we estimated *Pdetection* as a *Pert*(0.7,0.9,1) distribution.

**Sensitivity of PCR testing.** Unlike the more common standard direct movements, all quarantine-based movement types have an element of risk mitigation using pre- and/or post-quarantine PCR testing. PCR sensitivity (*PCR_Se*) was modelled as a *Beta*(9.65,1.19) distribution as previously described [19]. For those pathways where infection could have occurred at the quarantine location: it is plausible that the timing of the PCR would not detect early incubating virus and its sensitivity estimate was halved to account for this.

**Pre-quarantine infection clearance.** For all quarantine-based movements, the period between the start of the journey, to the equid being released from quarantine may exceed the period of viraemia in incubating and post-incubating equids. Horse movement associated quarantine is a minimum of 14-days, and closer to 16 days when allowing for pre-release sample processing and testing. In this period horses that enter the pathway with a probability of infection can potentially clear infection if they are subclinical and do not die. (We assume AHS related clinical presentation with or without death in quarantine would result in investigation and detection, as per protocol [7]). To account for this, a binary *clearance* factor was included in the model. The likelihood of clearance relates to the incubation and infectious period of AHSV, a concept used in similar models [18, 20]. The highest risk scenario is where equids are infected on the day of entry into quarantine, with viraemia persisting until after the quarantine period is over. To establish the probability of clearance of extrinsically infected equids (i.e., outside of quarantine) we sum estimates of the incubation period and viraemic period to obtain an overall *riskperiod*. Incubation periods for horses were modelled by *Pert* (2,6,10) [18] and infectious period by *Gamma*(29.75, 4.95) [20]. A random number of days prior to entry to quarantine, to allow for variation in the time when equids may have been infected prior to entry, was then added to a conservative total quarantine period of 14 days. A range of zero to 30 days was used to reflect the monthly intervals used through the analysis. This figure gives an effective period (*periodtoclear*) which, if exceeded by the *riskperiod* results in a *clearance* factor of one. Alternatively, clearance is likely reflected by a *clearance* factor of

zero. Natural clearance is possible for equids entering quarantine and being infected during quarantine. This was not, however, considered in the model and all equids that potentially were infected during quarantine remained infectious after exit. For zebra movements the infectious period was amended to reflect the likely extended infectious period for this species to a *Gamma*(9.502, 0.411) distribution. This distribution was based on experimental data [21] and was fitted using maximum likelihood estimation [22]. The incubation period for zebra remained as for horses. While the expected period at risk for zebra did increase, the period of quarantine for these animals is 21 days, which is also reflected as such in the analysis.

**Further considerations for quarantine-based movements.** For all non-direct, i.e. quarantine based movements, there are two non-mutually exclusive pathways for infection to occur–equids can be infected prior to quarantine and/or equids can be infected during quarantine. The probability of infection at these two points may vary. In the first instance origin data prior to quarantine for all but the *VPSOQ* movements were not available, and therefore *P_inf* needed to be estimated for these pathways. To do this, median *P_inf* rates for all local municipalities were established for each month of the year. A subset of these rates was extracted to represent those areas where both cases had been reported and movements had originated at any point during the year. The median of these median *P_inf* rates was then identified per month and used as the probability of infection for these risk pathways ($P_{inf\_est\_origin}$). In the second instance, *P_inf* during quarantine was estimated from the location where quarantine took place. For *VPSOQ* movements we assume zero probability of infection while undergoing quarantine as quarantine takes place in the AHS protection zone of the controlled area and no cases of AHS have been reported in the protection zone since 2014 OIE WAHID–accessed October 2020. As a result the *VPSOQ* movements have only a single pathway of infection.

Risk processes are summarised in Table 2. Individual pathways are highlighted where applicable. Outcomes for each movement class are the pathway probability of at least 1 undetected infection for *n* equids moved during period *i* from each local municipality in the country. Local municipality and month probability of entry were aggregated to establish final monthly probability of entry of AHSV into the AHS controlled area. Monthly probabilities of entry were also aggregated to obtain an annual median probability of entry.

## General considerations

One horse moved from the south of Namibia via a *VPSOQ* movement. The underlying population at risk, and hence probability of infection, could not be established for this region of origin. The movement occurred in September 2019. Namibia reported AHS cases in the country in 2019, with the last case reported in June 2019, approximately 650 km north of the movement origin. The only reported cases of AHS in Namibia were recorded in January 2019, approximately 1450 km north of the origin of movement (OIE WAHID–accessed October 2020). The origin of the movement in question was within 5 km of the border of South Africa and was allocated to the nearest local municipality in South Africa (GID 184) to ensure inclusion in the model.

## Sensitivity and 'what-if' analysis

A sensitivity analysis was performed to establish parameter uncertainty in relation to the conditional means of the primary model output. Input distributions were fixed at a lower and upper limit of 1% and 99% respectively. The model output used was the probability of entry of AHSV for each month at local municipality level (i.e., prior to aggregation for all areas) from areas and during months where movement occurred. Furthermore, based on results, sensitivity analysis was performed on standard direct and standard SOQ movements only. Standard

**Table 2. Risk processes for the five movement pathways for equine movement into the African horse sickness controlled area of South Africa.**

| Movement class | Pathway | Pathway probability of at least 1 undetected infection for $n$ equids moved during period $i$ from area $l$ |
|---|---|---|
| 1. Standard direct | Overall | $1 - (1 - [(P_{inf_{origin}} \times P_{subclin}) + [P_{inf_{origin}} \times (1 - P_{subclin}) \times (1 - P_{detection})]])^n$ |
| 2. Standard stop-over quarantine | 1: infection prior to quarantine | $Clearance \times ([P_{inf\_est\_origin} \times (1 - SePCR) \times P_{subclin}] + [P_{inf\_est\_origin} \times (1 - SePCR) \times (1 - P_{subclin}) \times (1 - P_{detection})])$ |
| | 2: infection during quarantine | $[P_{inf\_soq} \times \left(1 - \frac{SePCR}{2}\right) \times P_{subclin}] + [P_{inf\_soq} \times \left(1 - \frac{SePCR}{2}\right) \times (1 - P_{subclin}) \times (1 - P_{detection})]$ |
| | Overall | $1 - (1 - (P_{SOQ_{pathway1}} + P_{SOQ_{pathway2}} - (P_{SOQ_{pathway1}} \times P_{SOQ_{pathway2}})))^n$ |
| 3. Zebra | 1: infection prior to quarantine | $Clearance \times [P_{inf\_est\_origin} \times (1 - SePCR)^2]$ |
| | 2: infection during quarantine | $P_{inf\_q} \times \left(1 - \frac{SePCR}{2}\right)$ |
| | Overall | $1 - (1 - (P_{Zeb_{pathway1}} + P_{Zeb_{pathway2}} - (P_{Zeb_{pathway1}} \times P_{Zeb_{pathway2}})))^n$ |
| 4. Vector protected quarantine | 1: infection prior to quarantine | $Clearance \times ([P_{inf\_est\_origin} \times (1 - SePCR)^2 \times P_{subclin}] + [P_{inf\_est\_origin} \times (1 - SePCR)^2 \times (1 - P_{subclin}) \times (1 - P_{detection})])$ |
| | 2: infection during quarantine | $[P_{inf\_vpqo} \times \left(1 - \frac{SePCR}{2}\right) \times P_{subclin}] + [P_{inf\_vpqo} \times \left(1 - \frac{SePCR}{2}\right) \times (1 - P_{subclin}) \times (1 - P_{detection})]$ |
| | Overall | $1 - (1 - (P_{VPQO_{pathway1}} + P_{VPQO_{pathway2}} - (P_{VPQO_{pathway1}} \times P_{VPQO_{pathway2}})))^n$ |
| 5. Vector protected stop-over quarantine in AHS controlled area | Overall | $Clearance \times ([P_{inf\_origin} \times (1 - SePCR)^2 \times P_{subclin}] + [P_{inf\_origin} \times (1 - SePCR)^2 \times (1 - P_{subclin}) \times (1 - P_{detection})])$ |

direct movements accounted for most of the risk of entry. Standard SOQ movements were included since, of the remaining movement types, it was the next most influential and included quarantine-based parameters such as the use of pre-movement testing using PCR, incubation and infectious periods associated with infection.

A baseline probability of entry was determined, in which no control measures considered in the model were applied. In this case probability of entry only considered probability of infection at the time of movement. A risk differential and reduction factor were established between the current risk of entry using control measures and the uncontrolled movement risk to establish the impact of current control measures.

### Data considerations

Spatial layers relating to local municipalities, associated provinces and South Africa's border are based on those available from the South African Demarcation Board where there are no limitations on use (https://www.arcgis.com/home/item.html?id=bbf702f73c7742749a64c68ac5324aa9 –accessed 2020-11-30). For integration with other datasets an edited version which is coded and can be linked to movement and case data is available at the data repository associated with this manuscript. The R code used to perform and depict the risk assessment is also included in a GitHub repository (https://github.com/UP-COP-SPSRA/ahsv_entry_assessment_zafcontrolledarea).

## Results

### Probability of infection

Monthly probability of infection was estimated for areas where cases were reported and/or from which movements of equids occurred. The maximum median probability of infection, at local municipality level, occurred primarily between February and May (Fig 2). Fig 3 depicts the spatial distribution of median probability of infection estimates, with each local municipality depicted by

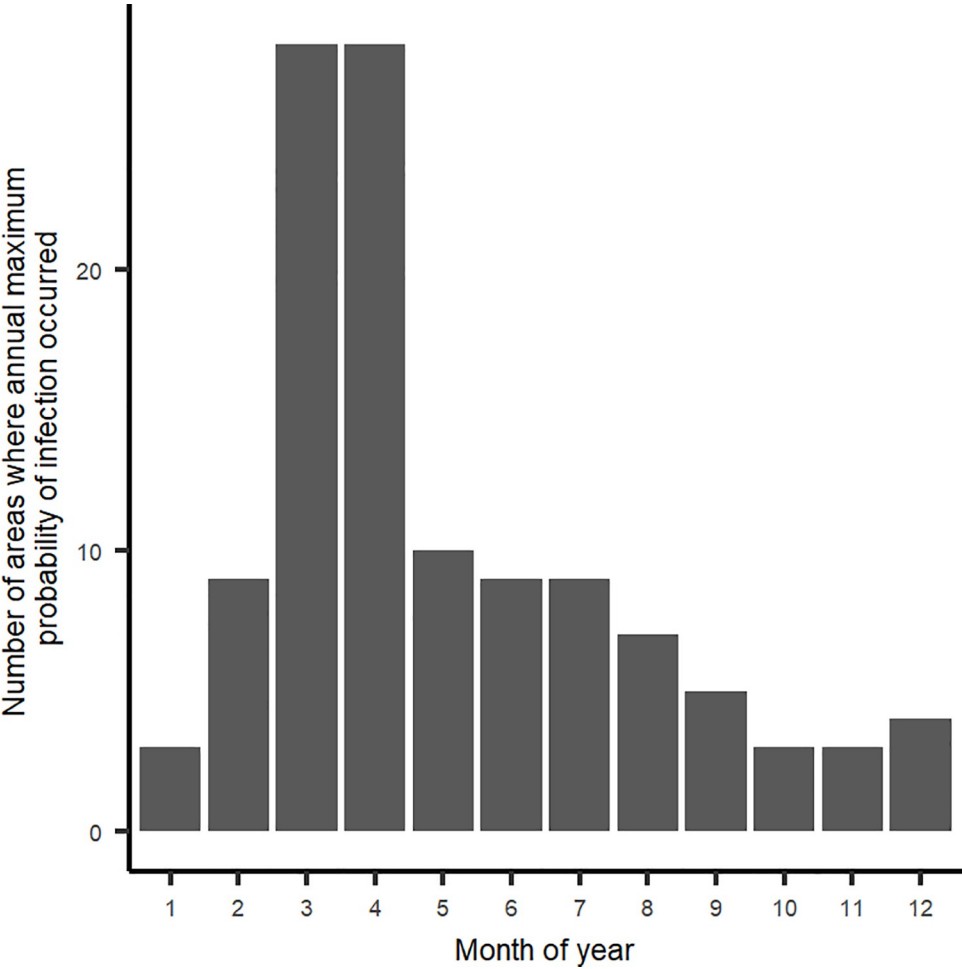

**Fig 2. Monthly count of local municipalities where the maximum median probability of infection of African horse sickness was estimated.**

the maximum level during the year. Those areas where probability of infection equalled or exceeded 1.5% are labelled by the month (numerical) in which that level occurred.

**Probability of entry.** The monthly probability of entry of AHSV into the AHS controlled area from all local municipalities is shown in Table 3 and Fig 4. The highest median probability of entry is present in February at a level of 5.73% (95% CI: 2.14%-15.72%). Probability of entry is relatively stable throughout the year except for February, and a slight rise from August through October (Fig 4). Overall cumulative probability of entry through all movement types for the year is estimated at 20.21% (95% CI: 15.89% - 28.89%)

The maximum median probability of entry by local municipality and month is shown in Fig 5. Generally, areas did not exceed a maximum median monthly probability of entry of > 0.1%. Areas where this is exceeded are labelled by the month (numerical) in which their maximum median probability of entry is at its greatest. The area and month of highest individual probability of entry was the Port Elizabeth region during February which had a median probability of entry of 3.89% (95% CI: 0.54% - 14.13%).

**Sensitivity analysis and 'what-if' scenario.** The probabilities of entry of AHSV through standard direct movements (Fig 6) and standard SOQ movements (Fig 7) at a local municipality level were both most sensitive to probability of infection–the former at the origin of movement,

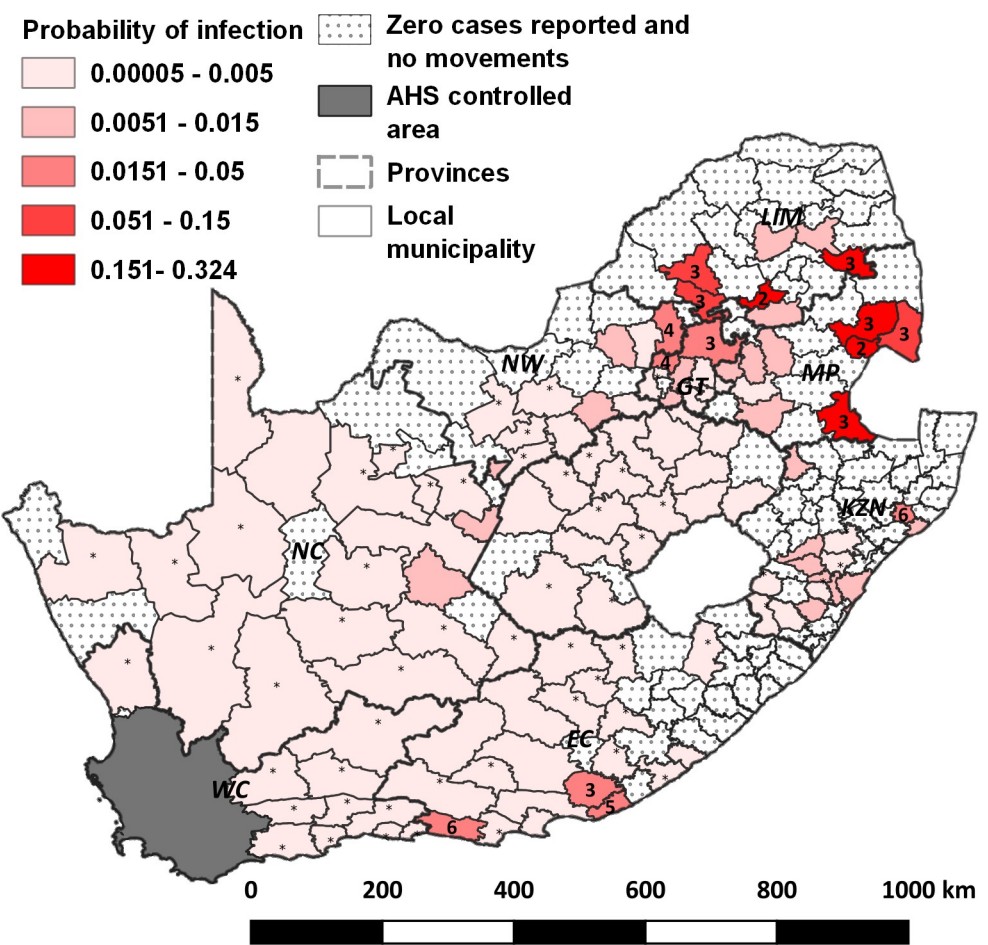

**Fig 3. Maximum median probability of infection of African horse sickness (AHS) virus by local municipality and month in South Africa.** Probability of infection is only depicted in areas where either cases were reported and/or where equine movements into the AHS controlled area (grey area in south-west part of the country) occurred. Areas where cases and movements did not occur are depicted by a dot fill. Areas where movements occurred from but where no cases were reported depicted by a star. Case months, for those areas where the maximum probability of infection was > = 1.5%, are labelled with the month number. Provinces of the country are shown by dashed lines and are labelled. WC = Western Cape; EC = Eastern Cape; KZN = KwaZulu-Natal; MP = Mpumalanga; LIM = Limpopo; NW = North West; GT = Gauteng; FS = Free State; NC = Northern Cape.

the latter at the location of the stop-over quarantine facility. Three additional parameters that could be evaluated through quarantine-based movements (i.e., incubation period, infectious period and the sensitivity of the PCR used for testing) all had relatively low impact on the overall variability in probability of entry (Fig 7) when compared to other parameters.

The 'what-if' scenario, where previously evaluated control measures are removed, resulted in an overall cumulative probability of entry of 56.96% (95% CI: 44.09%-74.95%) with an overall percentage change in risk between control and non-control of 63.90% (95% CI: 50.22%-74.53%). Monthly risk decrease resulting from control ranged between 38.30% in October to 94.70% in January (1.7 and 18.9 reduction factors respectively—Table 4).

## Discussion

### Outcomes

Median monthly probability of entry of AHSV into the AHS controlled area ranged from 0.75% (June) to 5.73% (February) for all movements into the AHS controlled area of South

**Table 3. Probability (median and 95% credibility interval) of the entry of African horse sickness virus (AHSV) into the AHS controlled area in South Africa through the legal movement of equids from the endemic area of the country.**

| Month | Probability of entry (median and 95% credibility interval) | | | | | |
|---|---|---|---|---|---|---|
| | All movements[a] | Standard direct movements | Standard stop-over quarantine | Vector protected quarantine | Vector protected stop-over quarantine in AHS controlled area | Zebra |
| January | 0.01156 (0.0078–0.01798) | 0.01155 (0.0078–0.01788) | | | 0 (0–0) | |
| February | 0.05734 (0.02142–0.15715) | 0.05536 (0.0198–0.15552) | | 0.00169 (0.00032–0.00586) | 0 (0–0) | |
| March | 0.01572 (0.00798–0.0325) | 0.01559 (0.00785–0.03235) | 0.00012 (0.00004–0.00033) | | | |
| April | 0.00921 (0.00437–0.02094) | 0.00705 (0.00297–0.01838) | 0.00019 (0.00007–0.00053) | 0.0016 (0.00033–0.00565) | 0 (0–0) | |
| May | 0.0145 (0.00717–0.02927) | 0.01183 (0.00504–0.02642) | 0.00207 (0.00094–0.00442) | 0.00033 (0.00005–0.00125) | 0 (0–0) | |
| June | 0.00745 (0.00357–0.01544) | 0.00642 (0.00293–0.01351) | 0.001 (0.00042–0.00245) | | 0 (0–0) | |
| July | 0.00979 (0.00589–0.01743) | 0.00864 (0.00499–0.016) | 0.00099 (0.00043–0.00225) | 0.00006 (0.00001–0.00022) | | |
| August | 0.01795 (0.01259–0.02661) | 0.01783 (0.01248–0.02648) | 0.00003 (0.00001–0.00011) | 0.00005 (0.00001–0.00017) | | 0.00003 (0.00001–0.00008) |
| September | 0.02136 (0.01428–0.03542) | 0.02133 (0.01424–0.0354) | 0.00003 (0.00001–0.00011) | | 0 (0–0) | |
| October | 0.0239 (0.01401–0.04746) | 0.02388 (0.01397–0.0474) | 0.00003 (0.00001–0.00011) | | | |
| November | 0.01084 (0.00717–0.0175) | 0.01074 (0.00708–0.01742) | 0.00006 (0.00001–0.00022) | 0.00002 (0–0.00005) | | |
| December | 0.01306 (0.00843–0.02039) | 0.01286 (0.00824–0.02016) | 0.0001 (0.00003–0.00029) | 0.00008 (0.00002–0.00027) | | |
| Annual cumulative | 0.20208 (0.15887–0.28887) | 0.19422 (0.1511–0.28193) | 0.00493 (0.00322–0.00783) | 0.00428 (0.00175–0.00981) | 0 (0–0.00007) | 0.00003 (0.00001–0.00008) |

Blank cells reflect where, for months and movement types, no movements took place. Values are rounded to 5 decimal places.

[a]All movements reflect the aggregation of the five different movement types.

Africa, with annual median probability of entry estimated at 20.21% (95% CI: 15.89–28.89%). To the best of our knowledge this is the first attempt to estimate risk of entry and quantify risk mitigation in an AHS endemic country where zoning principles are applied to control the disease and establish a free area. It is therefore not possible to directly compare these estimates with similar studies. In the 20 years between 1999 and 2019, two outbreaks of AHS have occurred that were introduced, through movement of an infected horse, into the AHS controlled area [10]. This corresponds to an annual probability of introduction by movement of ~10%. Our evaluation was that of the potential entry of AHSV into the AHS controlled area and does not imply introduction, which is the establishment and the spread into at least one controlled area resident equid. de Vos et al. and Faverjon et al., albeit in different environmental and movement contexts, found a three- and six-times factor differential between probability of entry versus introduction of AHSV through movement of equines into The Netherlands and France respectively [18, 23]. We believe our estimate of ~20% median annual probability of entry reflects a reasonable and conservative estimate when compared to the known occurrence of introduction through the movement of equids in South Africa.

Sergeant et al. quantified the probability of exporting AHS undetected infected horses from South Africa [19]. Included in that analysis was the evaluation of horses exported from the

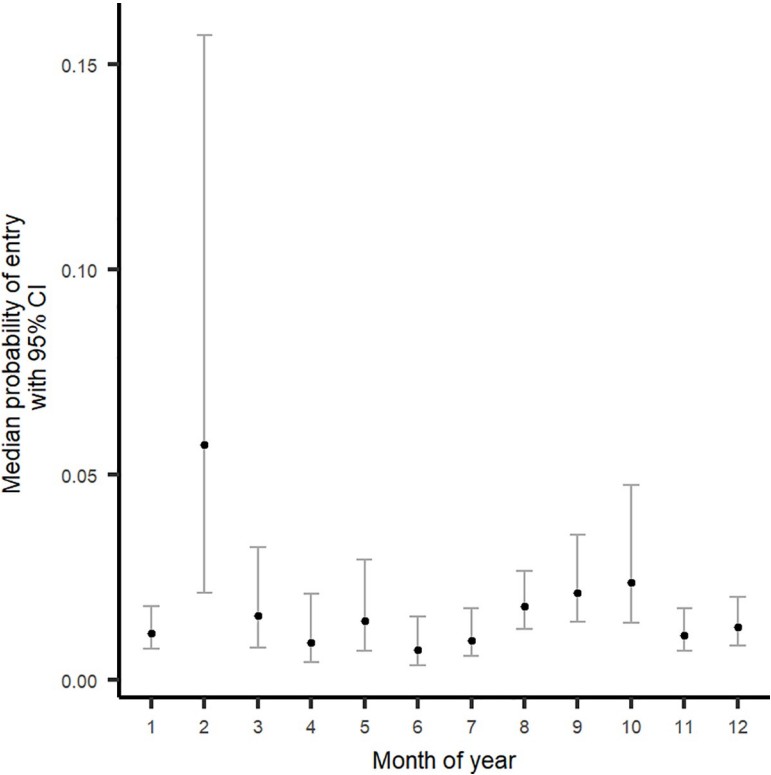

**Fig 4. Monthly probability (median and 95% credibility interval) of the entry of African horse sickness virus (AHSV) into the AHS controlled area in South Africa through the legal movement of equids from the endemic area of the country.**

endemic/infected part of South Africa through a similar process to VPQO. Our annual estimate of risk associated with this pathway is approximately 2.2 times that of Sergeant et al., but well within their estimated 95% predictive estimates. While not substantial, this difference is likely due to the higher resolution of probability of infection estimates from our data. Sergeant et al. effectively used cumulative incidence across the estimated population in the endemic area. In our estimates three of the seven months where VPQO movements occurred had reported case totals, resulting in higher estimates of probability of infection than the cumulative incidence for the country in those months.

We could find two publications that relate to the quantitative entry risk assessments of AHSV into known free countries (The Netherlands and France) [18, 23]. De Vos et al. considered international horse movements only and found median probabilities of entry into the Netherlands between 0.003% (February) and 0.03% (July) with an annual median probability of entry of 0.15%. The overall annual median probability of entry in France [23] was 0.3% for an infectious host (they had also considered movement of infected vectors) while median entry risk varied between 0.026% to 0.095% in the second half of the year. These estimates are substantially lower than those we estimate for movements within South Africa. The primary reasons for this are likely that import requirements for equids into Europe are very restrictive from AHS high-risk countries, and very few horses are likely to move in this fashion. Faverjon et al. indicate that equine hosts arriving from high-risk countries made up only 1.6% of a total of ~1600 animals moved per year. In our analysis the use of an accumulated monthly incidence, as an input into probability of infection in those areas where zero cases were reported, effectively means that all movements would be considered high risk in the European import

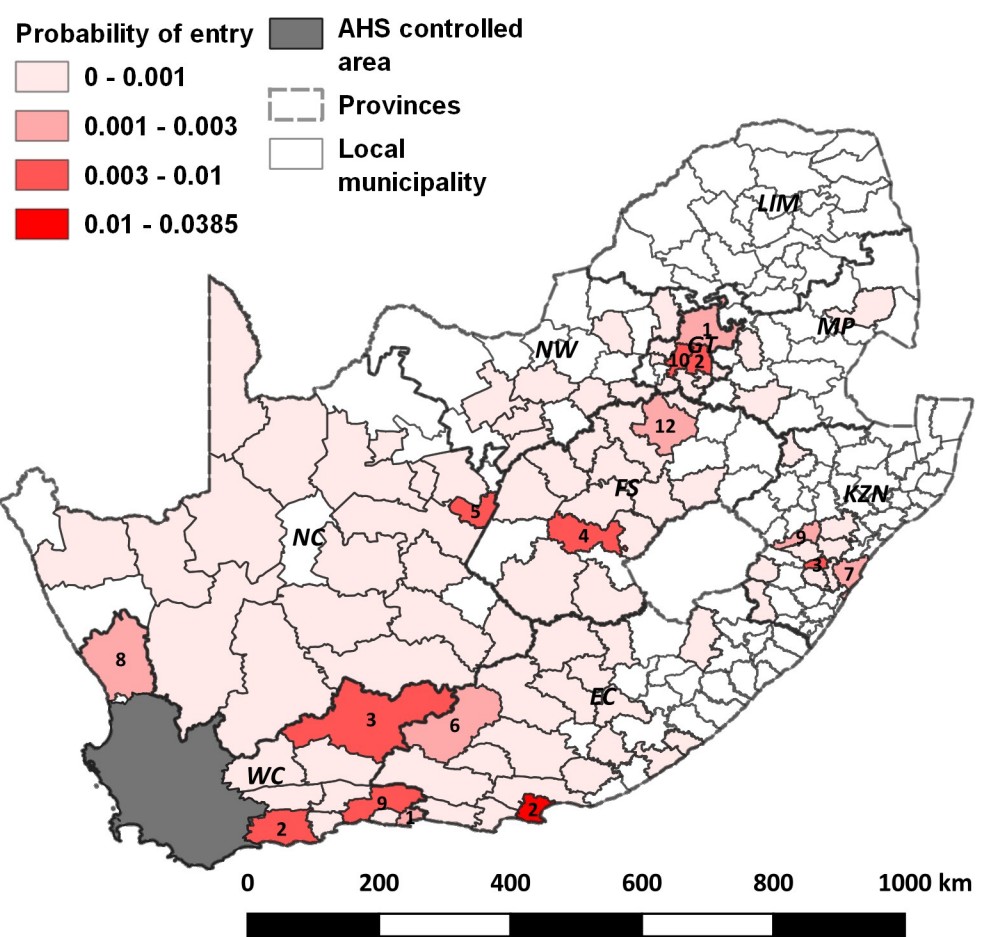

**Fig 5. Maximum median probability of entry of African horse sickness virus (AHSV) into the AHS controlled area by local municipality and month.** Case month for those areas where the probability of entry was > = 0.001 are labelled with the month number. Areas with no colour fill are those from where no movements took place. Provinces of the country are shown by dashed lines and are labelled. WC = Western Cape; EC = Eastern Cape; KZN = KwaZulu Natal; MP = Mpumalanga; LIM = Limpopo; NW = North West; GT = Gauteng; FS = Free State; NC = Northern Cape. The AHS controlled area is shown in grey, located in the south-western part of the country.

context. It is interesting in that both de Vos et al.'s and Faverjon et al.'s estimates at seasonal level were largely inverse compared to that of our analysis. Their highest risk was distinctly in the second half of the year. This is likely to be from difference in movement patterns within South Africa compared to imported horses into the Netherlands and France. Faverjon et al. explicitly indicate the seasonal risk is due to importation patterns and associated risk profiles of equine origins differing during the year.

While probability of infection was generally highest in the north-eastern parts of South Africa, movement patterns resulted in entry risk from most of South Africa except for the far north-eastern region. The primary periods of entry risk were in February (late summer) and again in late winter/early summer. The latter finding is important–while late summer is high risk considering the seasonal nature of the disease, late winter and early summer is not a period when AHS risk is perceived to be high [8], and our findings show that entry risks should not be underestimated during this time. Furthermore, the spatial distribution of entry risk highlights that, while movement and case patterns may be well defined, collaboration between all provincial regulatory stakeholders is important and must be maintained throughout the year,

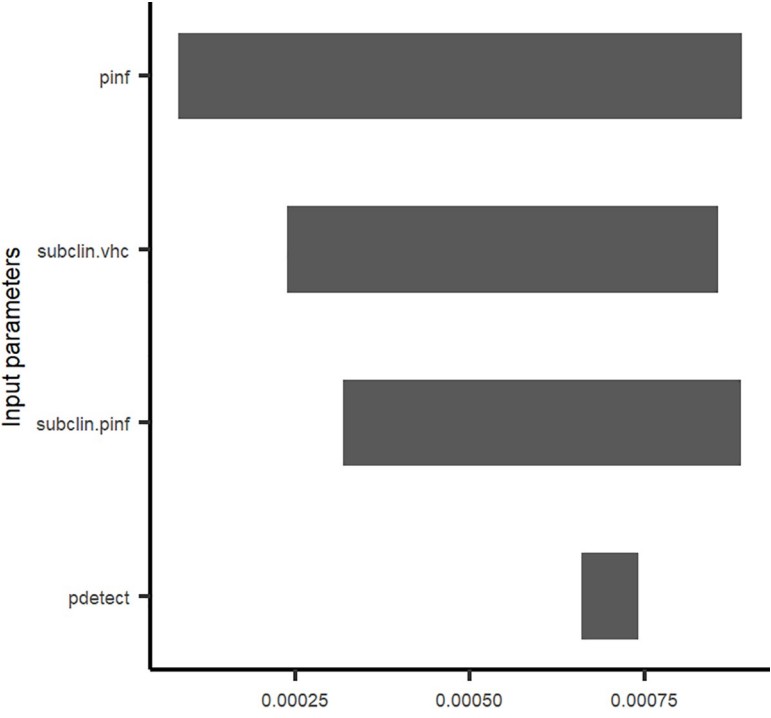

**Fig 6. Tornado plot showing variability in the probability of entry of African horse sickness virus (AHSV) into the AHS controlled area, at local municipality level and for standard movements, as a result of model parameters.** Variability is depicted by impact of parameters on the conditional mean of the probability of entry. pinf = probability of infection at origin; subclin.vhc = estimate of the sub-clinical rate of AHS cases in relation to infected horses not being clinically ill (and therefore undetected, during pre-movement veterinary health certification; subclin. pinf = estimate of the sub-clinical rate of AHS cases used to modulate case number estimates for each local municipality and month of movement; pdetect = probability veterinarians detect clinically ill AHS infected horses during veterinary health certification.

not just during the late summer classical 'AHS season'. Also of interest is that the Western Cape Province (WCP) local municipalities that are not in the AHS controlled area play an important role in risk of entry of AHSV. The last case reported prior to 2020 of AHS in the WCP was in May 2016 (OIE WAHIS–accessed 8 December 2020), and in general cases from this province are sporadic except in years when outbreaks in the controlled area have occurred (OIE WAHIS–accessed 8 December 2020). It is important that this sporadic case occurrence does not lead to complacency in terms of health certification of horses, as many of the movements into the AHS controlled area originate from within the WCP itself. The importance of sensitisation to clinical signs and detecting cases was perhaps best illustrated by the Port Elizabeth local municipality in the Eastern Cape. Here, even though the use of a partial AHS risk declaration was implemented (see below), a single case detected during February resulted in the highest median municipal level probability of entry of 3.89% for that month where one case occurred, and 47 horses moved.

The movement type most influencing overall risk of entry of AHSV was standard direct movements from the AHS endemic area into the AHS controlled area. This is not unexpected, but it highlights that quarantine associated movements, while perceived to be high risk and undertaken with more scrutiny and risk mitigation, play a lesser role in entry risk. This also implies that those parameters that influence standard direct movement risk are critical in

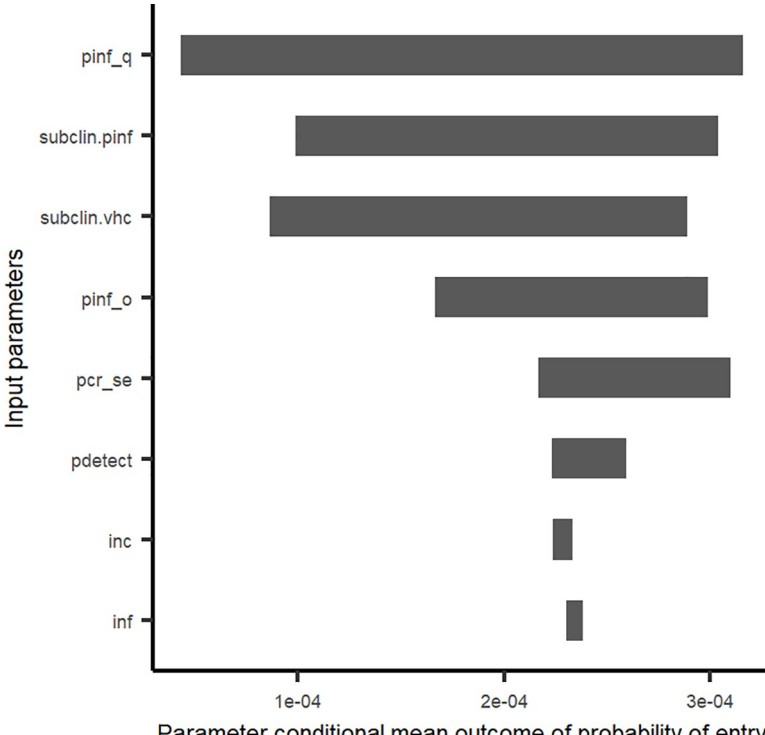

**Fig 7. Tornado plot showing variability in probability of entry of African horse sickness virus (AHSV) into the AHS controlled area, at local municipality level and for standard stopover quarantine movements, as a result of model parameters.** Variability is depicted by impact of parameters on the conditional mean of the probability of entry. pinf_q = probability of infection during quarantine; subclin.pinf = estimate of the sub-clinical rate of AHS cases used to modulate case number estimates for each local municipality and month of movement; subclin.vhc = estimate of the sub-clinical rate of AHS cases in relation to infected horses not being clinically ill (and therefore undetected, during pre-movement veterinary health certification; pinf_o = probability of infection prior to entry into quarantine; pcr_se = sensitivity of the PCR test used prior to exit from quarantine; pdetect = probability veterinarians detect clinically ill AHS infected horses during veterinary health certification; inc = incubation period estimate for AHSV infection; inf = infectious period of AHSV infection.

mitigating overall risk. In this light the role of private veterinarians in identifying and reporting cases of AHS in the endemic area, which impacts probability of infection, and their role in clinical health certification, is an important aspect of control that requires continued attention.

Subclinical rates resulted in relatively high uncertainty in entry risk. Subclinical rates were obtained from studies of outbreaks in the AHS controlled area as similar studies were not available in large populations in the AHS endemic area. More informative data from the endemic area, where annual AHS vaccination is mandatory, may provide narrower estimates on subclinical rates in horses due to be moved into the AHS controlled area and provide a clearer indication of consequent entry risk.

The 'what-if' scenario, where control is removed, shows the positive impact of control measures currently in place. While there are some additional measures taken when issuing permits which are not included in this evaluation, but which would also decrease entry risk, such as identification of horses and ensuring that compliant AHS vaccination has taken place, the overall impact of control decreases annual risk of entry by ~ 64%. Expressed as a proportion change this reflects a 2.8 times decrease in entry risk when control is implemented. January had a high and variable risk differential and reduction. Further investigation showed that, while movements and case numbers were relatively low compared to other months, three

**Table 4. Outcome of the 'what-if' scenario where the probability of entry of African horse sickness virus (AHSV) into the AHS controlled area was modelled with no control measures in place during movement.**

| Month | Probability of entry (median and 95% credibility interval) | | Entry risk percentage change through control (median percentage and 95% credibility interval) | Entry risk reduction factor through control (median times and 95% credibility interval) |
|---|---|---|---|---|
| | All controlled movements | All uncontrolled movements | | |
| January | 0.01156 (0.0078–0.01798) | 0.21956 (0.05896–0.53173) | -0.94696 (-0.98009–0.79501) | 18.9 (4.9–50.2) |
| February | 0.05734 (0.02142–0.15715) | 0.10504 (0.04597–0.23376) | -0.40357 (-0.70597–0.21079) | 1.7 (1.3–3.4) |
| March | 0.01572 (0.00798–0.0325) | 0.02684 (0.01521–0.04985) | -0.38702 (-0.63112–0.23833) | 1.6 (1.3–2.7) |
| April | 0.00921 (0.00437–0.02094) | 0.09204 (0.04814–0.187) | -0.89838 (-0.96325–0.75334) | 9.8 (4.1–27.2) |
| May | 0.0145 (0.00717–0.02927) | 0.10613 (0.05177–0.22044) | -0.86236 (-0.94515–0.68724) | 7.3 (3.2–18.2) |
| June | 0.00745 (0.00357–0.01544) | 0.03395 (0.01666–0.07286) | -0.77726 (-0.91548–0.51964) | 4.5 (2.1–11.8) |
| July | 0.00979 (0.00589–0.01743) | 0.01802 (0.01198–0.02956) | -0.44046 (-0.64717–0.26507) | 1.8 (1.4–2.8) |
| August | 0.01795 (0.01259–0.02661) | 0.03022 (0.02272–0.04169) | -0.39793 (-0.52997–0.29381) | 1.7 (1.4–2.1) |
| September | 0.02136 (0.01428–0.03542) | 0.0359 (0.02601–0.05518) | -0.39104 (-0.56353–0.27683) | 1.6 (1.4–2.3) |
| October | 0.0239 (0.01401–0.04746) | 0.04048 (0.02539–0.07219) | -0.38302 (-0.60168–0.25037) | 1.6 (1.3–2.5) |
| November | 0.01084 (0.00717–0.0175) | 0.01829 (0.01305–0.02787) | -0.39745 (-0.5572–0.2825) | 1.7 (1.4–2.3) |
| December | 0.01306 (0.00843–0.02039) | 0.02206 (0.01575–0.03247) | -0.40199 (-0.56047–0.28426) | 1.7 (1.4–2.3) |
| Annual cumulative | 0.20208 (0.15887–0.28887) | 0.56964 (0.44094–0.74951) | -0.63902 (-0.74532- -0.50218) | 2.8 (2–3.9) |

Outcomes of the overall probability of entry are shown for reference. A percentage difference and risk reduction factor between controlled and uncontrolled movements is depicted with 95% credibility intervals.

movements occurred (through vector protected stop-over quarantine in the AHS controlled area) from a single area which had a median probability infection of 7.31% (95% CI: 1.31% - 21.83%). This resulted in higher estimates and the large variability reflects the variability in the probability of infection at origin. This highlights that low numbers of high-risk movements can have a big impact on risk, even when most risk is from standard direct movements under strict control measures.

There are uncertainties regarding the entry risk of AHSV through the legal movement of equids. While we have approached the estimation of this risk conservatively, the key findings are not the actual risk levels. Of more importance is the identification of space-time patterns or movement parameters that influence risk which may be used to develop further control strategies and policy, or at least to provide an idea of when vigilance against potential hazards is most necessary. There are other pathways of AHSV entry into the AHS controlled area. The use of live attenuated AHS vaccine in the area is a known risk which has resulted in outbreaks [10]. There is also the risk of infected vector introduction, either through the transport of animals (both equines and non-equines) or through wind dispersal mechanisms; although there is no evidence that transport or wind dispersal of infective vectors has played a role in previous outbreaks of AHS in the AHS controlled area.

## Data and parameter considerations

A single calendar year of movement and case data was selected to estimate important model parameters. Movement data for 2019, when compared to a 12-month period between September 2017 and August 2018, did not differ substantially temporally, spatially or by number of equids moved [16]. A total of 613 cases made up the 2019 case dataset which is well above the average of previous years. Between 2014 and 2018 (5 years) the Department of Agriculture, Land Reform and Rural Development (DALRRD - http://dalrrd.gov.za/ - accessed 8 December 2020) reported, on average, 236 AHS cases across the country per year, with most reported in 2014 (389 cases). The epidemiology of the disease is largely driven by host population, vector population and climatic/environmental factors [1] and the spatial extent of the disease is unlikely to differ substantially from year to year in the AHS endemic area. Since 1993 (i.e. in the last 26 years) DALRRD has reported more AHS cases than in 2019 in a calendar year on only four occasions– 1127 cases in 2001, 1027 cases in 2011, 904 cases in 2008, and 675 cases in 2013.

There were several features of control that were not considered that may have resulted in a conservative overestimation of entry risk. For quarantine-based movements we did not consider the reduction of infection risk which has been included in similar assessments where reduction of infection risk during quarantine was estimated as 50–90% [18, 23, 24]. The primary reason for excluding this is that quarantine in standard SOQ movements, which were the majority of quarantine-based movements, are not vector protected to the same extent that VPQO movements are, and quarantine is used effectively to reclassify the status of a horse based on PCR testing while it remains in a low-risk area prior to release. For those movements that use vector protected quarantine, there are still periods where horses are allowed outside of vector protection between 10:00 and 15:00 (based on the crepuscular nature of *Culicoides*). While estimates of the probability of vector protection breakdown in South African export quarantine facilities have been used to estimate the protective nature of quarantine [19], vector count data from vector protected facilities used for movement into the AHS controlled area of South Africa is not available, and estimates would be speculative. A further potentially protective factor of quarantine that was not considered is that daily temperature monitoring of horses is mandatory [7], and clinical cases can be detected in this fashion.

There were limited zebra moved during the evaluated period. While zebra movement conditions are highly restrictive, if movement volumes of this species increase, a re-evaluation of this pathway would be warranted.

Standard direct movements are only permitted if the area of origin is classified as low risk by the responsible Government veterinarian. This risk at origin is recorded as an Area Status Declaration (ASD) [7]. ASD based risk levels were not explicitly considered in the model as this risk profile, if considered high, would prevent standard direct movements from an entire State veterinary region. This is accounted for in the analysis by using totals of equids moved to estimate entry risk. There is however a partial ASD AHS risk classification used to demarcate areas directly surrounding (~30 km) sporadic cases which allows direct movements from outside these restricted areas. This would therefore result in months where direct movements may have occurred, and where cases are reported for the same local municipality. We did not incorporate this aspect of control as estimates of risk differentials using a buffer around positive cases with a vector borne disease are unknown. While outbreaks in the AHS controlled area have generally been spatially limited [9, 10], wind-based dispersal of potentially infected AHSV midges has been documented [25, 26].

In previous risk assessment work on AHS in South Africa [19] an underreporting factor along with a case fatality rate factor was used to improve case total estimates in the AHS

endemic area. In that case, however, deaths were generally the primary case type notified. In our evaluation we use both laboratory and clinical based data with amplification by an estimated subclinical rate, and we expect that through this mechanism our estimates of case totals are likely to accurately reflect the reality. To provide conservative estimates of AHSV entry risk through standard direct movements we also assumed that infection is synonymous with potential viremia. The setting of *Pinf* to 0.5 where case estimates were greater than the underlying population did not influence model outcomes. This occurred in 9 iterations of determining *Pinf* and occurred in months and from areas where movement did not occur. Finally, de Vos et al. estimated the clinical probability of detection at 70% during pre-movement health certification [18], which was the lower bound of our estimate (*Pert*(0.7,0.9,1)). In the context of South Africa we believe that, since pre-movement veterinary health certification is specifically aimed at AHS detection, and that the veterinarians performing pre-movement health certification are sensitised to AHS, our estimate is realistic and justifiable.

## Conclusion

We provide a spatial-temporal risk assessment of the probability of entry of AHSV into the AHS controlled area of South Africa through the legal movements of equids using 2019 data as a basis. The median annual risk of entry of AHSV is 20.21% (95% CI: 15.89%-28.89%). This value will vary with changing underlying infection rates and disease extent in the endemic area, but the data used for this estimate is likely to be conservative due to the above average number of cases used in determining the underlying probability of infection at local municipal level. Standard direct movements result in the highest probability of entry, and spatial and temporal evaluation show that, while seasonal risks of entry are highest in late summer and late winter/early summer, location of risk is present in much of the country. This includes the Western Cape Province which has historically reported only sporadic cases of the disease. This evaluation, and the sensitivity analysis thereof, show that vigilance by regulatory and private veterinarians forms the most important aspect of entry risk mitigation through equine movements. The low volumes of highly regulated quarantine-based movements play a negligible role in overall risk. Further understanding of subclinical rates of disease in the endemic area of South Africa will decrease the uncertainty of risk estimation. Finally, control measures currently in place are likely to decrease the entry risk of AHSV into the AHS controlled area through legal equine movements by 2.8 times.

## Acknowledgments

South African horse census data, used to modulate published raster-based data, was provided by the DALRRD. We thank the Western Cape Department of Agriculture for access to movement and area status declaration data, and the SAEVA for the collaboration in managing the AHS case reporting data. Much of the ECOD data is obtained from official testing laboratories and we are grateful to these laboratories and to DALRRD who liaise with SAEHP personnel to ensure that case totals can be readily obtained. We are grateful to Dr. Gary Buhrmann for kindly reviewing this manuscript.

## Author Contributions

**Conceptualization:** John D. Grewar.

**Data curation:** John D. Grewar, Camilla T. Weyer.

**Formal analysis:** John D. Grewar.

**Investigation:** Camilla T. Weyer.

**Methodology:** John D. Grewar, Johann L. Kotze.

**Software:** John D. Grewar.

**Supervision:** Camilla T. Weyer.

**Validation:** Johann L. Kotze.

**Visualization:** John D. Grewar.

**Writing – original draft:** John D. Grewar.

**Writing – review & editing:** John D. Grewar, Johann L. Kotze, Beverly J. Parker, Lesley S. van Helden, Camilla T. Weyer.

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
