## [Decision Letter · Decision Letter 0]

14 Apr 2021

PONE-D-21-06494

An exposure risk assessment of African horse sickness virus into the controlled area of South Africa through the legal movement of equids

PLOS ONE

Dear Dr. Grewar,

Thank you for submitting your manuscript to PLOS ONE. After careful consideration, we feel that it has merit but does not fully meet PLOS ONE’s publication criteria as it currently stands. Therefore, we invite you to submit a revised version of the manuscript that addresses the points raised during the review process.

The reviewer recommends that you make substantial revisions to your manuscript. I therefore request you to attend to all the concerns that were raised by the reviewer and resubmit your revised manuscript as advised in this letter.

We look forward to receiving your revised manuscript.

Kind regards,

Martin Chtolongo Simuunza, PhD

Academic Editor

PLOS ONE

Journal Requirements:

1. Please ensure that your manuscript meets PLOS ONE's style requirements, including those for file naming. The PLOS ONE style templates can be found athttps://journals.plos.org/plosone/s/file?id=wjVg/PLOSOne_formatting_sample_main_body.pdf and https://journals.plos.org/plosone/s/file?id=ba62/PLOSOne_formatting_sample_title_authors_affiliations.pdf

'SAEHP functions through public private partnership agreement with the Western Cape

Department of Agriculture: Veterinary Services. The ECOD case numbers used in this

evaluation reflect the reporting system numbers and do not necessarily reflect the

official totals as reported by South Africa’s Veterinary Services.'.

Additional Editor Comments (if provided):

Reviewers' comments:

Reviewer's Responses to Questions

**Comments to the Author**

1. Is the manuscript technically sound, and do the data support the conclusions?

Reviewer #1: Yes

2. Has the statistical analysis been performed appropriately and rigorously? 

Reviewer #1: N/A

3. Have the authors made all data underlying the findings in their manuscript fully available?

Reviewer #1: Yes

4. Is the manuscript presented in an intelligible fashion and written in standard English?

Reviewer #1: Yes

5. Review Comments to the Author

Reviewer #1: An exposure risk assessment of African horse sickness virus into the controlled area of

South Africa through the legal movement of equids

Line 54 and 55: The disease occurs in both domestic (horses and donkeys) and wild (zebra) equine hosts.

Kindly use scientific words for the names of animals at first mention and then in brackets for common names

Line 83-84: The sentence ‘In this study we estimate, using a stochastic model, the probability of exposure (OIE described as entry assessment)’ is confusing. Should it be ‘OIE described as entry assessment or OIE described entry and exposure assessment as you are estimating the probability of exposure. The OIE risk assessment process has 4 components; release/entry assessment, exposure assessment, consequence assessment and risk estimation. From the sentence above it is not clear if you are estimating the probability of release or of exposure.

In my opinion, I would like to point out that what was conducted was a release assessment and thus this should reflect in the title, objectives and other areas of the article and the article can be published as such.

For an exposure assessment, further parameters need to be imputed in the model such as:

Population of horses in the controlled area, probability of transmission to susceptible horses, probability of detection in the controlled area e.t.c.

6. PLOS authors have the option to publish the peer review history of their article (what does this mean?). If published, this will include your full peer review and any attached files.

Reviewer #1: No

---

## [Author Response · Author response to Decision Letter 0]

21 Apr 2021

We’d like to thank the reviewer for the time they took to review this manuscript, it is improved and become more accurate based on their suggestions.

Reviewer comment #1: Line 54 and 55: The disease occurs in both domestic (horses and donkeys) and wild (zebra) equine hosts. Kindly use scientific words for the names of animals at first mention and then in brackets for common names

Response #1: Thank you for the comment – that section has been reworded as follows:

“The disease occurs in Equus caballus (domestic horse), Equus africanus asinus (domestic donkey) and zebra (of which Equus quagga (common zebra) and Equus zebra (mountain zebra) are the most prevalent species in South Africa). Domestic horses are most susceptible to the disease and are the most likely of the hosts to show overt clinical signs [1,2].”

Reviewer comment #2: Line 83-84: The sentence ‘In this study we estimate, using a stochastic model, the probability of exposure (OIE described as entry assessment)’ is confusing. Should it be ‘OIE described as entry assessment or OIE described entry and exposure assessment as you are estimating the probability of exposure. The OIE risk assessment process has 4 components; release/entry assessment, exposure assessment, consequence assessment and risk estimation. From the sentence above it is not clear if you are estimating the probability of release or of exposure.

In my opinion, I would like to point out that what was conducted was a release assessment and thus this should reflect in the title, objectives and other areas of the article and the article can be published as such. For an exposure assessment, further parameters need to be imputed in the model such as: Population of horses in the controlled area, probability of transmission to susceptible horses, probability of detection in the controlled area e.t.c.

Response #2: Thank you for this very important clarification. We agree completely with the reviewer and have made changes to the manuscript throughout to reflect rather an ‘entry’ risk assessment rather than an exposure risk assessment. While this resulted in multiple changes throughout the manuscript, they were effectively editorial. The description of reviewed manuscripts in the discussion (Faverjon and de Vos et al.) were also referred to as entry risk assessments components to standardise this concept and bring in line with OIE naming. Lastly the title of the manuscript has been changed to:

“An entry risk assessment of African horse sickness virus into the controlled area of South Africa through the legal movement of equids”

The manuscript code in the repository has been reviewed to reflect this change as well – a new repository on GitHub was made so that the repository name also reflected this change.

#Response to editors: The changing of the naming of the risk assessment from exposure to entry risk did require figures 4 through 7 to have labels changed and these have been included in the re-submission. The request for editorial changes to formatting have been made. I’m not sure if there was an issue with the short title being included on page one? All figures were processed with PACE and the newly attached figures are those that were reformatted by PACE.

Regarding the request for clarification on competing interests: I can confirm that the authors have declared that no competing interests exist. It wasn’t clear where I should include this in the online system but please let me know if it is required in places it is omitted from.

Thank you for your efforts

John Grewar on behalf of the authors

---

## [Decision Letter · Decision Letter 1]

11 May 2021

An entry risk assessment of African horse sickness virus into the controlled area of South Africa through the legal movement of equids

PONE-D-21-06494R1

Dear Dr. Grewar,

We’re pleased to inform you that your manuscript has been judged scientifically suitable for publication and will be formally accepted for publication once it meets all outstanding technical requirements.

Kind regards,

Martin Chtolongo Simuunza, PhD

Academic Editor

PLOS ONE

Additional Editor Comments (optional):

Reviewers' comments:

Reviewer's Responses to Questions

**Comments to the Author**

1. If the authors have adequately addressed your comments raised in a previous round of review and you feel that this manuscript is now acceptable for publication, you may indicate that here to bypass the “Comments to the Author” section, enter your conflict of interest statement in the “Confidential to Editor” section, and submit your "Accept" recommendation.

Reviewer #1: All comments have been addressed

2. Is the manuscript technically sound, and do the data support the conclusions?

Reviewer #1: Yes

3. Has the statistical analysis been performed appropriately and rigorously? 

Reviewer #1: N/A

4. Have the authors made all data underlying the findings in their manuscript fully available?

Reviewer #1: Yes

5. Is the manuscript presented in an intelligible fashion and written in standard English?

Reviewer #1: Yes

6. Review Comments to the Author

Reviewer #1: I am happy that you took time to address my comments raised in the previous round of review. Therefore, I have no further comments and that the manuscript can be published.

7. PLOS authors have the option to publish the peer review history of their article (what does this mean?). If published, this will include your full peer review and any attached files.

Reviewer #1: No

---

## [Editor Report · Acceptance letter]

17 May 2021

PONE-D-21-06494R1 

An entry risk assessment of African horse sickness virus into the controlled area of South Africa through the legal movement of equids 

Dear Dr. Grewar:

I'm pleased to inform you that your manuscript has been deemed suitable for publication in PLOS ONE. Congratulations! Your manuscript is now with our production department. 

Kind regards, 

on behalf of

Dr. Martin Chtolongo Simuunza 

Academic Editor

PLOS ONE